# Risk factors for Encapsulating Peritoneal Sclerosis in patients undergoing peritoneal dialysis: A meta-analysis

**Dashan Li, Yuanyuan Li, Hanxu Zeng, Yonggui Wu** [ORCID] *

Department of Nephropathy, the First Affiliated Hospital, Anhui Medical University, Hefei, Anhui, China

* wuyonggui@medmail.com.cn

**Data Availability Statement:** All relevant data are within the paper and its Supporting Information files.

## Abstract

### Purpose

Encapsulating Peritoneal Sclerosis (EPS) is the most serious complication of long-term peritoneal dialysis (PD), which considerably reduces the patient's quality of life, leading to patients discontinuing PD. Considering these negative effects, it is necessary to systematically review and determine the risk factors of EPS.

### Methods

The PubMed, Embase, Web of Science, Cochrane Library, and China Biology Medicine (CBM) were searched from their inception to January 1st, 2022, and the bibliographies from the citations of relevant articles were manually searched. The ROBINS-I (Risk of Bias in Non-randomized studies of Interventions) tool was used to evaluate the risk of bias of included studies.

### Results

Ten studies involving 12595 participants were included in this meta-analysis. The results revealed that a younger age at PD onset (MD = -7.70, 95% CI, -11.53~-3.86), a higher transporter (MD = 0.13, 95% CI, 0.09~0.18), a longer PD duration (SMD = 1.15, 95% CI, 0.68~1.61), a longer peritonitis duration (MD = 12.66, 95% CI, 3.85~21.47), and history of glomerulonephritis (OR = 1.42, 95% CI, 1.02~1.97) were significant risk factors for EPS. However, sex, use of icodextrin, the number of peritonitis episodes, and history of multicystic kidney disease did not affect the risk of EPS.

### Conclusions

This review provides a scientific basis for further understanding the etiology of PD-related EPS and improving prevention strategies. More high-quality studies are necessary to validate this paper's findings.

**Funding:** The author(s) received no specific funding for this work.

**Competing interests:** The authors have declared that no competing interests exist.

## Introduction

Encapsulating Peritoneal Sclerosis (EPS), also known as Peritoneal Fibrosis and Sclerosing Encapsulating Peritonitis, is a rare but the most severe complication associated with peritoneal dialysis, especially in patients who are on long-term PD [1]. In 2000, the International Society for Peritoneal Dialysis has described EPS as a rare clinical syndrome characterized by intermittent, persistent or recurrent intestinal obstruction caused by diffuse adhesions of a thickened and sclerosing peritoneum [2]. If obstruction does not resolve, intestinal mural ischemia and peritoneal adhesions may further develop, resulting in infections, bloody dialysate, systemic inflammatory response syndrome, sepsis, and death [3]. Adverse outcomes of PD-related EPS are diverse, including loss of incipient peritoneal ultrafiltration capacity, a higher transport status, and a lower residual kidney function [4]. Although the underlying mechanism by which normal peritoneal membrane progresses to peritoneal fibrosis in PD patients is not yet clear, overexpression of a cascade of proinflammatory cytokines (e.g. transforming growth factor β1, interleukin-6) and profibrotic gene (e.g. Fibronectin 1, α smooth muscle actin), neoangiogenesis, and mesothelial to mesenchymal transition (MMT) may be involved in the occurrence and progression of EPS [5–7]. Unfortunately, the optimal treatment for EPS is not clear. There are no standard therapies or randomized controlled trials to cure EPS patients but some empiric knowledge based mainly on observational evidences include surgical and medical interventions, might be beneficial [8]. Generally, PD should be discontinued after the diagnosis of EPS is made. Additional therapeutic strategies include steroid therapy and other immunosuppression, nutritional support and assessment, tamoxifen, parenteral or enteral nutrition, and peritoneal lavage [8, 9]. Despite these, the mortality rates of patients with EPS are still 25%-55% in adults [10–12], with positively correlated with the duration of PD therapy.

The overall incidence of EPS obtained from a variety of countries is low and varies between 0.5% and 7.3% [13–18]. A registry study from Australian reported that the incidence rate increased with duration of PD therapy [14]. Similar results were obtained in a prospective study from Japan, in which the incidence of EPS was 0.7%, 2.1%, 5.9%, 17.2% in patients who had been on PD for more than 5, 8, 10, and 15 years, respectively [13]. Many factors including duration of PD, age at initiation of PD, bio-incompatible dialysate, high-transporter membrane, duration of UFF (ultrafiltration failure), duration of peritonitis, number of peritonitis episodes, and kidney transplantation may lead to EPS [1, 17, 19, 20]. However, both a single-center retrospective study from Taiwan and dual-center retrospective study from Iran analysed the relationship between age at PD onset and the development of EPS, with the latter concluding that younger age was a high-risk factor for EPS [21], while the former found no significant difference according to age [22]. The question, therefore, 'What are the risk factors related to EPS in PD patients?' remains unanswered.

Regarding controversial results of existing studies, a meta-analysis was conducted to identify comprehensive results which could contribute to develop clinical strategies for early prevention of EPS. To our knowledge, this is the first meta-analysis to analysis the risk factors associated with EPS.

## Methods

This meta-analysis was reported in line with PRISMA guidelines [23] and registered with the International Prospective Register of Systematic Reviews (PROSPERO) [24] with registration number (CRD42022302786).

### Search strategy

On 10 January 2022, two independent reviewers (Yuanyuan Li and Hanxu Zeng) searched relevant citations in PubMed, Embase, Web of Science, Cochrane Library, and

China Biology Medicine (CBM) from their inception to January 1st, 2022, and the bibliographies from the citations of relevant articles were manually searched. We also searched the http://www.greylit.org/ website for studies that were registered as completed but not yet published. There were no restrictions on language with limited to human subjects. We used a combination of search medical subject heading (MeSH) related to EPS and risk factor. Any discrepancies regarding study selection were resolved by additional assessment by a third investigator (Yonggui Wu). S2 File outlined the detailed search strategy of PubMed.

## Inclusion and exclusion criteria

The criteria for original research to be included in this meta-analysis were as follows: (1) This review included case-control studies, cross-sectional studies cohort studies, and randomized controlled trials (RCTs) to explore the risk factors associated with EPS, (2) The studies were conducted in patients whose peritoneal dialysis lasted three months at least and age >18years, (3) Availability of complete data required for the analysis, and (4) Clear and consistent diagnosis of EPS based on the diagnostic criteria. The exclusion criteria were (1) Reviews, conference papers, lectures, drug experiments, animal studies; (2) Studies that included duplicated data; (3) Primary data could not been extracted.

## Data extraction and risk assessment

Two authors (Yuanyuan Li and Hanxu Zeng) searched and extracted the data of the included primary studies, aggregating the resulting data into a structured table: first author's name, study duration, study type, country/region, proportion of women, sample size, and risk of bias. The ROBINS-I (Risk of Bias in Non-randomized studies of Interventions) tool was used by two authors (Yuanyuan Li and Hanxu Zeng) to evaluate the risk of bias of included studies based on the following seven domains: bias due to confounding; bias due to selection of participants; bias in classification of interventions; bias due to deviations from intended interventions; bias due to missing data; bias in measurement of outcomes; bias in selection of the reported result. This tool categorizes the risk of bias as low, moderate, serious, critical, and unclear [25]. Disagreements were resolved by consultation with a third reviewer (Yonggui Wu).

## Statistical analysis

All statistics were collated and analyzed using the Review Manager 5.3 and Stata 14.0 software. Continuous variables were summarized using mean difference (MD) or standardized mean difference (SMD) with their 95% confidence intervals (CI), For dichotomous variables, odds ratio (OR) with their 95% CI were estimated. In addition, heterogeneity was quantified using the Q test and $I^2$ statistics. When $I^2$ was < 50% and Q chisquared test result > 0.1, it shows that there is no large heterogeneity among the trials and the fixed-effect model was used; else a random-effect model was used. When large heterogeneity was present, sensitivity analysis were performed to identify responsible outlier studies. Furthermore, publication bias was further evaluated using the Egger's test. A trial sequence analysis (TSA) was used to analyze the sample size required for this meta-analysis to improve the credibility of this study.

SPSS Statistics version 25.0 (SPSS INC., Chicago, IL) was used to analyze the data of kappa of agreement during literature search. The data of kappa >0.60 were considered to indicate good agreement.

## Results

### Search results and study characteristics

A total of 495 potentially relevant articles were retrieved initially, in which 371 remained after duplicates from different databases were deleted. Next, 339 articles not aligned with the inclusion criteria were excluded after reading the titles and abstracts. Subsequently, 32 papers meeting the inclusion criteria were considered, in which 22 were excluded due to duplicate records, unadjusted confounding factors and inconsistent research design. Ultimately, we included 10 standards-compliant articles (Fig 1). The data of kappa of agreement during the systematic searches were 0.721, indicating good agreement.

Eight retrospective case-control studies and two prospective cohort studies [13, 15, 21, 22, 26–31] included in this meta-analysis were published between 2004 and 2018, with a total of 12595 patients on PD, which included 293 patients with EPS and 12302 with no EPS. Excluding Yamamoto et al.'s study [30], the number of participants in other studies ranged from 384 to 7618, and the incidence of EPS ranged from 0.4% to 11.7%. Study duration varied from 4 years to 35 years. In addition, two out of 10 studies had a women predominant (range of 52–53%). These studies were conducted in Iran, Japan, Netherlands, Taiwan, Australia, New Zealand, Turkey, and Ireland. The diagnosis of EPS in these ten articles was established based on the guidelines developed by the ISPD including clinical diagnosis, radiologic diagnosis and pathologic diagnosis. The basic characteristics of the selected articles are presented in Table 1. Potential bias was evaluated by the ROBINS-I tool for observational studies. All studies presented a moderate overall risk of bias, in which confounding bias remains in all studies due to their non-randomized nature. The exact details of the risk evaluation are summarized in S1 Table.

### Synthesis of the results

More than three studies involving identically defined risk factors for EPS were pooled in the meta-analysis. A total of 9 potential risk factors including age at PD onset, sex, duration of PD, dialysate-to-plasma creatinine ratio (D/P Cr), number of peritonitis episodes, duration of peritonitis, use of icodextrin, history of glomerulonephritis (GN), and history of polycystic kidney disease (PKD) were identified. The details are shown in Table 2. Heterogeneity was found in potential risk factors of exposure to age at PD onset, duration of PD, D/P Cr, peritonitis duration, and use of icodextrin. Therefore, the combined effect sizes of these factors were calculated using a random-effects model.

### Demographic factors for the occurrence of EPS

**Age at PD onset.** All studies that met the inclusion criteria reported on the relationship between age and EPS in a total of 10894 patients (311 patients with EPS and 10583 patients without EPS). Due to the obvious heterogeneity ($P < 0.001$, $I^2 = 82\%$) (Fig 2A), the random-effects model was applied, and the final results showed that significant statistical difference was found between age at PD onset and the development of EPS (MD = -7.70, 95% CI, -11.53~-3.86, $P < 0.0001$).

**Sex.** A fixed-effect model was utilized to analyze the data from 9 articles ($P = 0.98$, $I^2 = 0\%$), with a total of 10279 patients (5645 male and 4634 female patients). The comprehensive results suggested that in terms of sex, no statistical difference was found between the EPS patients and non-EPS patients (OR = 1.16, 95% CI, 0.89–1.52, $P = 0.26$) (Fig 2B).

**D/P Cr.** Three articles [28–30] provided reliable data to study the level of D/P Cr in a total of 888 patients (112 EPS and 776 non-EPS patients). Since the data analysis results showed

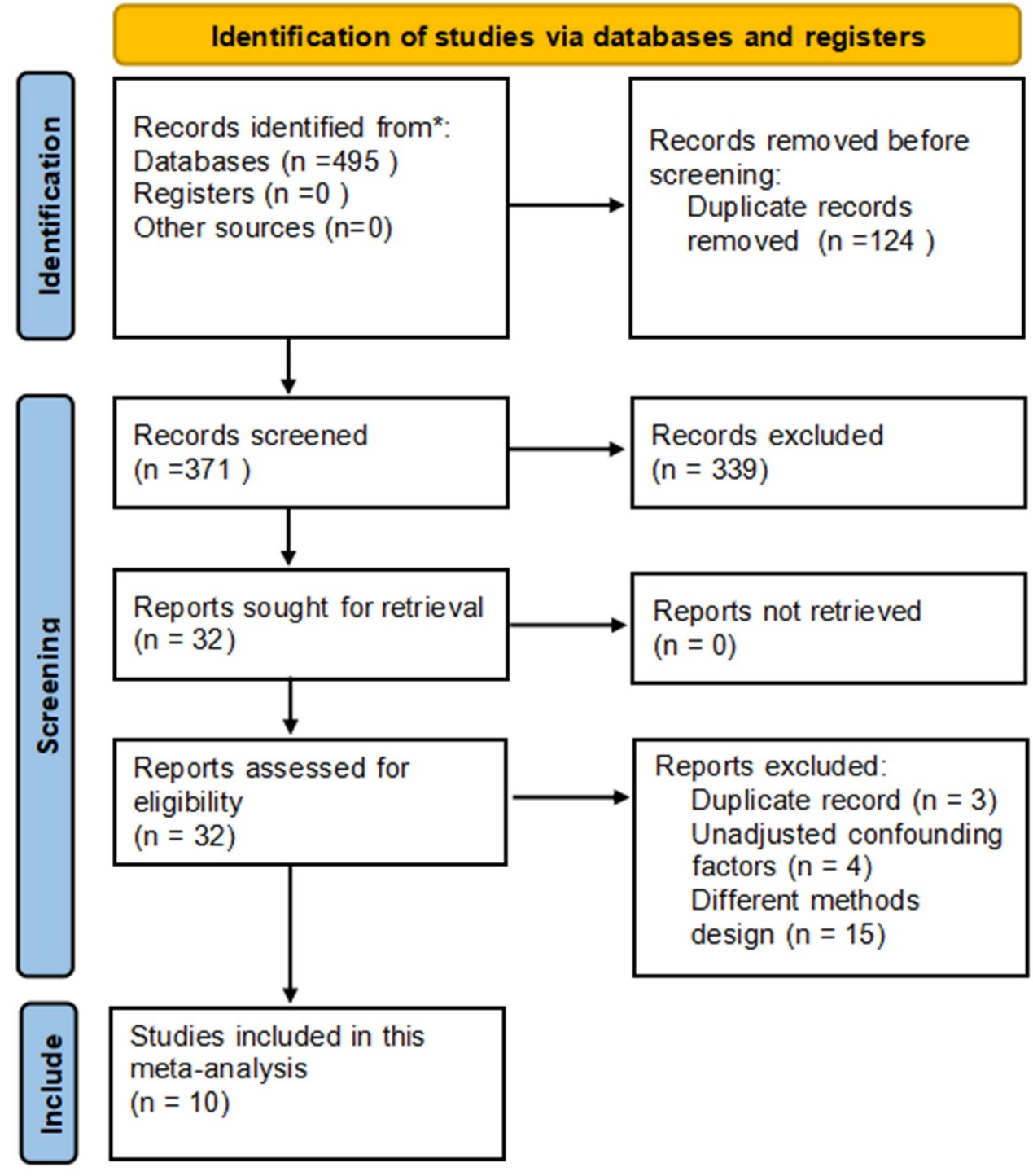

**Fig 1. Flow diagram of the literature selection in this meta-analysis.**

considerable heterogeneity (P = 0.03, $I^2$ = 72%), a random-effects model was used to analyze the combined MD with a consequence of 0.13 (95% CI, 0.09–0.18, P < 0.001) (Fig 2C).

**Duration of PD.** Seven articles [13, 21, 22, 26, 27, 29, 30] reported the relevant data, with a total of 1958 patients (223 EPS and 1735 non-EPS patients), among which we found

**Table 1. Basic characteristics of all studies in this meta-analysis.**

| Author | Study duration | Study design | Country/Region | Women(%) | Age range | | Sample sizes | | Risk of bias[a] |
|---|---|---|---|---|---|---|---|---|---|
| | | | | | EPS | NO EPS | | | |
| Alatab, S. [21] | 1995–2012 | Case control | Iran | 53 | 32.75±10.8 | 49.61±16.2 | case:12 | control: 122 | Moderate |
| Nakao, M. [29] | 1989–2010 | Case control | Japan | 33 | 54.6±11.5 | 56.8±13.0 | case:50 | control:57 | Moderate |
| Nakao, M. [28] | 1980–2015 | Case-control | Japan | 27 | 44±12 | 55±15 | case:44 | control:659 | Moderate |
| Yamamoto, R. [30] | NR | Case control | Japan | 33 | 48.9±12.8 | 52.6±10.4 | case:18 | control:60 | Moderate |
| Korte, M. R. [27] | 1996–2007 | Prospective cohort | Netherlands | 34 | 34.7±15.4 | 51.5±14.7 | case:63 | control:126 | Moderate |
| Hsu, H. J. [22] | 1990–2014 | Case control | Taiwan | 44 | 54.3±18.27 | 53.7±17.6 | case:6 | control:559 | Moderate |
| Johnson, D. W. [15] | 1995–2007 | Case control | Australia and New Zealand | 49 | 49.1±12.8 | 58.0±16.6 | case:33 | control:7585 | Moderate |
| Koc Y. [26] | 2001–2016 | Case control | Turkey | 52 | 40.8±14.6 | 45.9±16.6 | case:26 | control:358 | Moderate |
| Kawanishi, H. [13] | 1999–2003 | Prospective cohort | Japan | 38 | 54.7±8.9 | 56.7±14.1 | case:48 | control:453 | Moderate |
| Phelan, PJ. [31] | 1989–2008 | Case control | Ireland | 42 | 43.3±14.4 | 50.2±14.1 | case:11 | control:604 | Moderate |

[a] Risk of bias was evaluated using ROBINS-I tool.

significant heterogeneity (P < 0.001, $I^2$ = 87%) (Fig 3A). Ultimately, the results of the random-effects model obtained showed that the risk of EPS was higher in patients who have experienced PD for a longer time (SMD = 1.15, 95% CI, 0.68–1.61, P < 0.001).

**Number of peritonitis episodes.** Three studies [21, 29, 30] reported the impact of peritonitis on EPS, with a total of 319 patients (80 EPS and 239 non-EPS patients). Due to the low heterogeneity, a fixed-effects model was used (P = 0.21, $I^2$ = 35%). Notwithstanding, the final results did not describe the correlation between the number of episodes of peritonitis and the occurrence of EPS (OR = 1.59, 95% CI, 0.66–3.84, P = 0.30) (Fig 3B).

**Peritonitis duration.** A random-effects model was applied to evaluate the impact of peritonitis duration on EPS (P = 0.11, $I^2$ = 54%), with a total of 1375 patients (100 EPS and 1275 non-EPS patients). Three articles [22, 28, 29] were analyzed, and there was clear evidence that in patients with PD, the duration of peritonitis was a high-risk factor influencing the occurrence and development of EPS (MD = 12.66, 95% CI, 3.85–21.47, P = 0.005) (Fig 3C).

**Use of icodextrin.** Analyzing the data obtained in 3 articles [27–29], with a total of 999 patients (321 experimental and 678 control group), a random-effects model was used because of obvious heterogeneity (P < 0.001, $I^2$ = 96%) (Fig 4A). The final results suggested no statistical difference in the combined effect size (OR = 1.09, 95% CI, 0.08–15.40, P = 0.95).

**Table 2. Meta-analysis of the pooled risk factors for EPS.**

| Risk factors | Studies | Model | Heterogeneity test I2(%) | P | OR/MD/SMD | 95% CI | P-value |
|---|---|---|---|---|---|---|---|
| Age on dialysis | 10 | Random | 82 | <0.001 | -7.70 | (-11.53~ -3.86) | <0.001 |
| Sex | 9 | Fixed | 0 | 0.98 | 1.16 | (0.89–1.52) | 0.26 |
| D/P Cr | 3 | Random | 72 | 0.03 | 0.13 | (0.09–0.18) | <0.001 |
| PD duration | 7 | Random | 87 | <0.001 | 1.15 | (0.68–1.61) | <0.001 |
| Peritonitis duration | 3 | Random | 54 | 0.11 | 12.66 | (3.85–21.47) | 0.005 |
| Peritonitis history | 3 | Fixed | 35 | 0.21 | 1.59 | (0.66–3.84) | 0.30 |
| icodextrin | 3 | Random | 96 | <0.001 | 1.09 | (0.08–15.40) | 0.95 |
| GN | 6 | Fixed | 34 | 0.18 | 1.42 | (1.02–1.97) | 0.04 |
| PKD | 6 | Fixed | 0 | 0.61 | 0.68 | (0.31–1.46) | 0.32 |

Abbreviations: D/P Cr, dialysate-to-plasma creative ratio; GN, glomerulonephritis; PKD, polycystic kidney disease; OR, odds ratio; MD, mean difference; SMD, standardized mean difference.

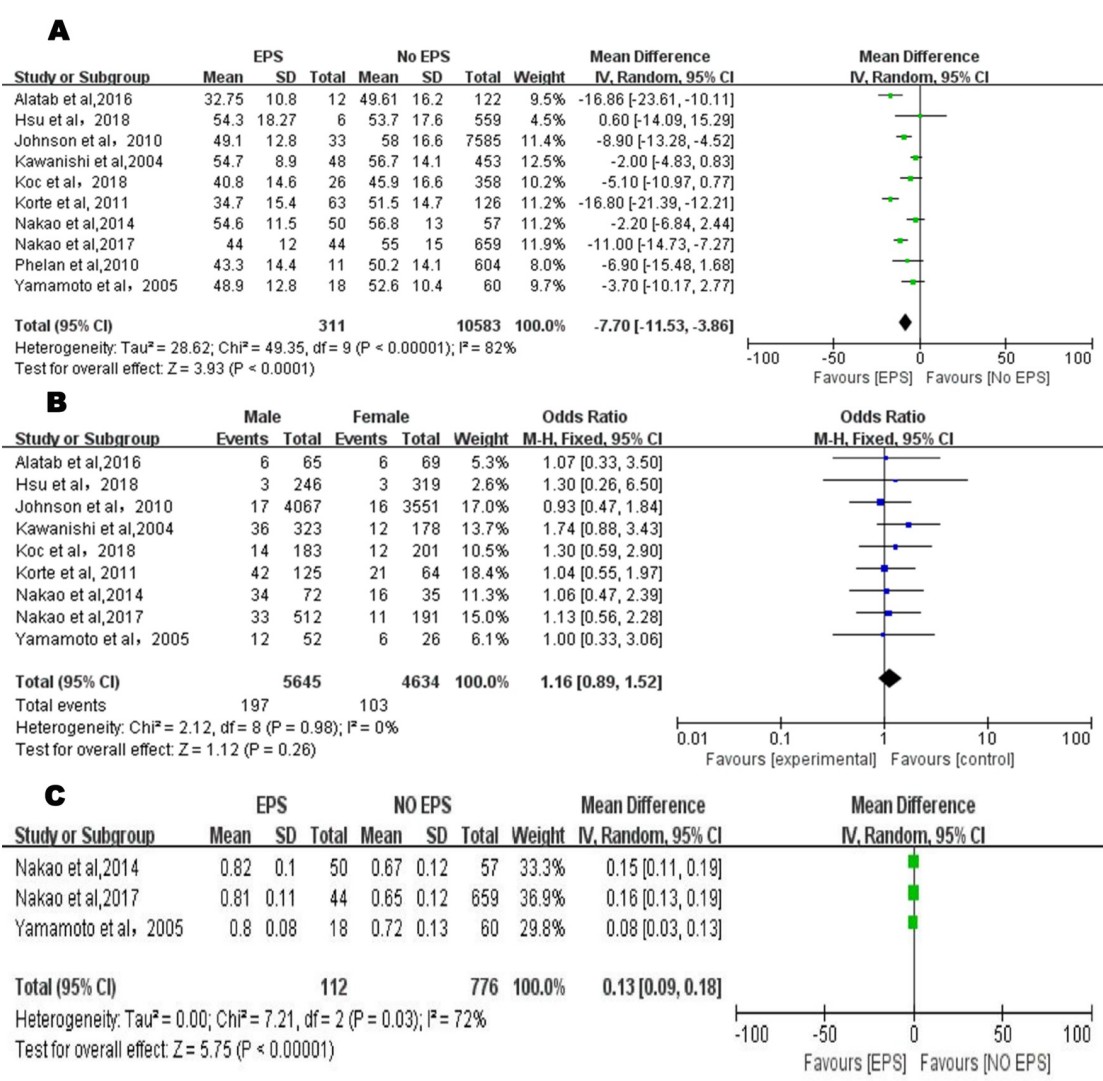

**Fig 2. A.** Forest plot of age at PD onset and EPS in the exploratory meta-analysis. The mean difference and 95% CI for each study and the final combined results are displayed numerically on the left and graphically as a forest plot on the right. **B.** Forest plot of sex and EPS in the exploratory meta-analysis. The odds ratio and 95% CI for each study and the final combined results are displayed numerically on the left and graphically as a forest plot on the right. **C.** Forest plot of D/P Cr and EPS in the exploratory meta-analysis. The mean difference and 95% CI for each study and the final combined results are displayed numerically on the left and graphically as a forest plot on the right.

**Glomerulonephritis (GN).** Six articles [15, 21, 26–29] reported a link between a history of glomerulonephritis and EPS, with a total of 9135 patients (2413 patients with glomerulonephritis and 6722 patients with no glomerulonephritis). A fixed-effects model was used for data analysis (P = 0.18, $I^2$ = 34%). The results suggested that patients with glomerulonephritis were more likely to develop EPS (OR = 1.42, 95% CI, 1.02–1.97, P = 0.04) (Fig 4B).

**Polycystic kidney disease (PKD).** After analyzing the data, 6 articles [15, 21, 26–29] were included in this analysis (P = 0.61, $I^2$ = 0%), with a total of 9135 patients (420 PKD patients and 8715 non-PKD patients). Subsequent results obtained using the fixed-effects model suggested that there was no statistical difference between the history of PKD and EPS (OR = 0.68, 95% CI, 0.31–1.46, P = 0.32) (Fig 4C).

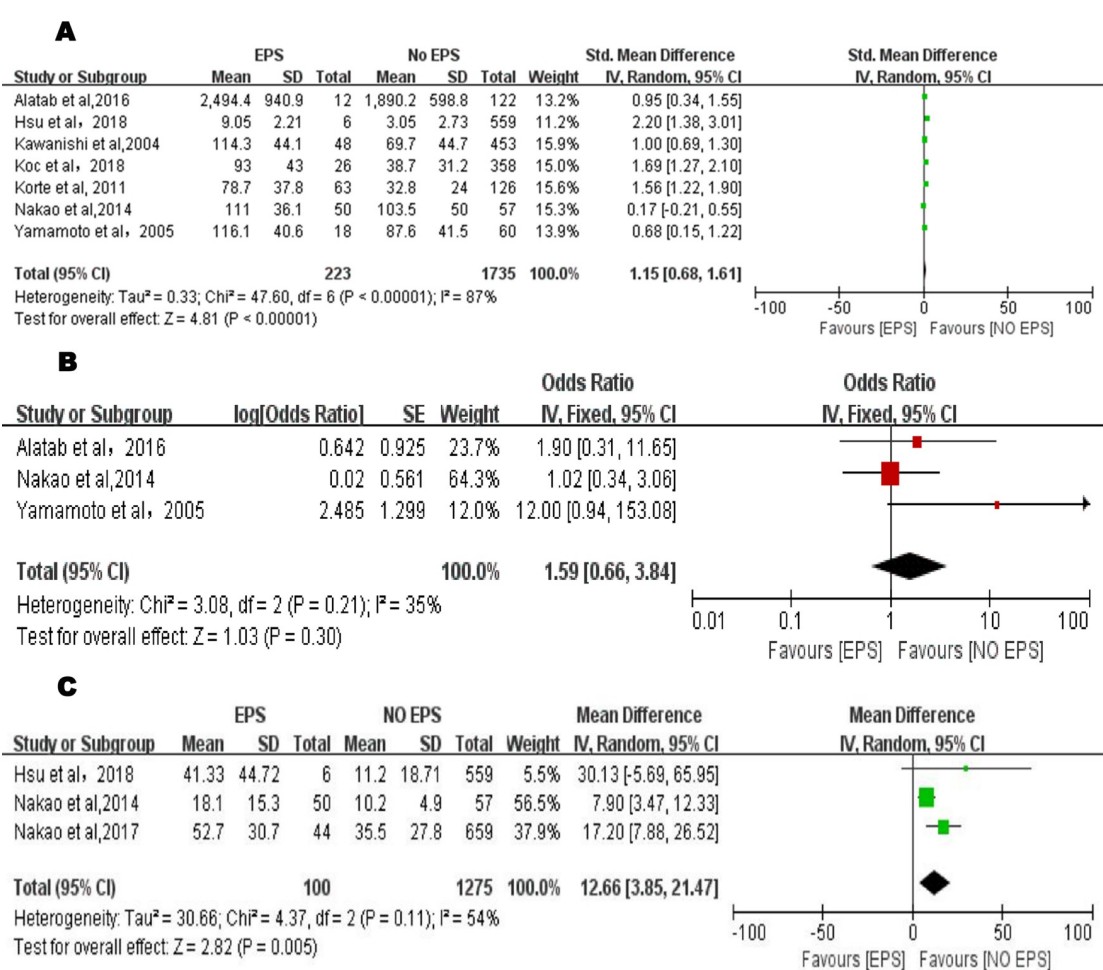

**Fig 3. A.** Forest plot of PD duration and EPS in the exploratory meta-analysis. The standard mean difference and 95% CI for each study and the final combined results are displayed numerically on the left and graphically as a forest plot on the right. **B.** Forest plot of number of peritonitis episodes and EPS in the exploratory meta-analysis. The odds ratio and 95% CI for each study and the final combined results are displayed numerically on the left and graphically as a forest plot on the right. **C.** Forest plot of peritonitis duration and EPS in the exploratory meta-analysis. The mean difference and 95% CI for each study and the final combined results are displayed numerically on the left and graphically as a forest plot on the right.

**Other risk factors.** By browsing these 8 selected studies, we found that serum creatine, serum calcium, serum phosphorus, serum albumin, hemoglobin, β2M (β2 microglobulin), systolic blood pressure, diastolic blood pressure, lupus erythematosus, APD (automated PD), HD (hemodialysis), fungus related peritonitis, CAD (coronary artery disease), etc, were mentioned in 2 studies. Furthermore, ducation, smoking history, history of dialysis, BMI (body mass index), weight, CVA (cerebrovascular accident), chronic lung disease, peripheral vascular disease, use of hypertonic solution, catheter site infection, immunoglobulin A nephropathy, renovascular disease, reflux nephropathy, hemolytic uremic syndrome, hypertension, collagen vascular disease, cancer, etc, were mentioned in one study. Considering the small number of studies involved and inadequate sample size, a large sample of clinical data is still necessary for further confirmation. Hence, the meta-analysis for these variables is not available here.

We performed a TSA of EPS-related risk factors, including age at PD onset, D/P Cr, peritonitis duration, and history of GN. Regarding the differences in data between different studies, the TSA on PD duration is not available here. As shown in S1 and S2 Figs, the Z-curves cross

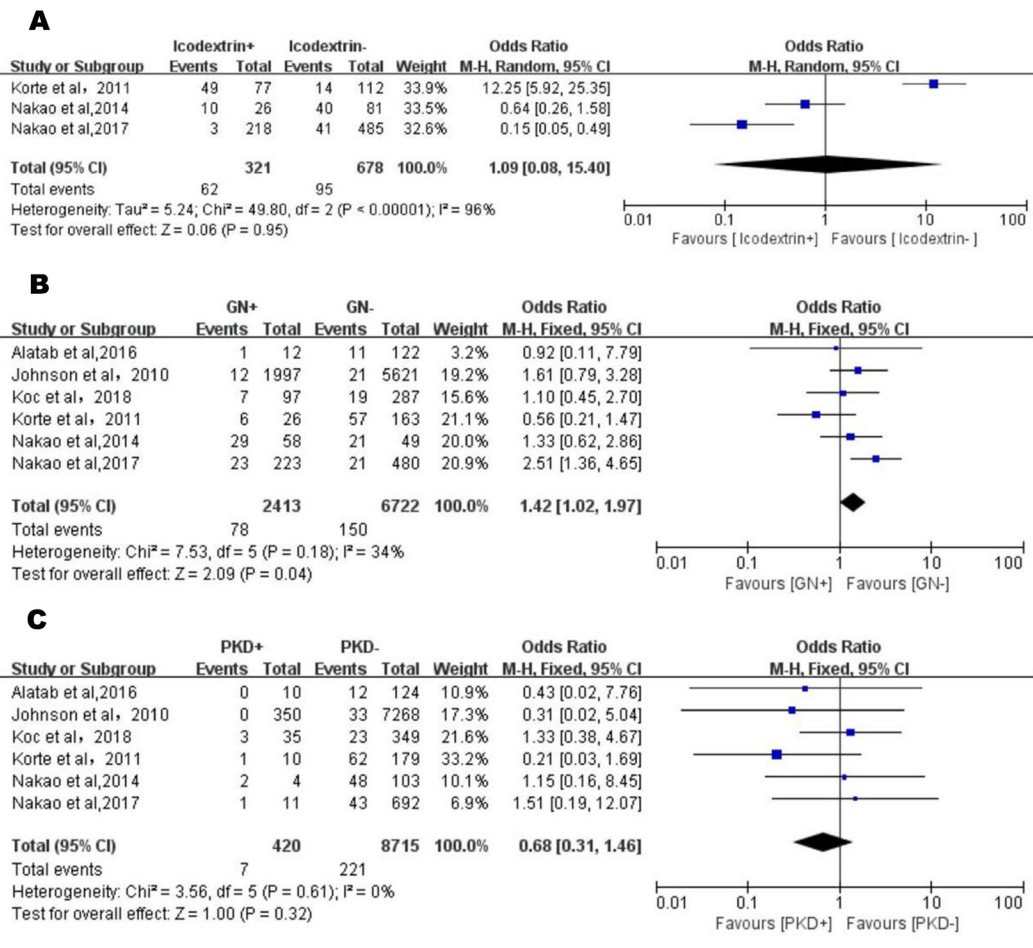

**Fig 4.** A. Forest plot of use of icodextrin and EPS in the exploratory meta-analysis. The odds ratio and 95% CI for each study and the final combined results are displayed numerically on the left and graphically as a forest plot on the right. B. Forest plot of GN and EPS in the exploratory meta-analysis. The odds ratio and 95% CI for each study and the final combined results are displayed numerically on the left and graphically as a forest plot on the right. C. Forest plot of PKD and EPS in the exploratory meta-analysis. The odds ratio and 95% CI for each study and the final combined results are displayed numerically on the left and graphically as a forest plot on the right.

both the conventional boundaries as well as O'Brien-Fleming alpha-spending boundaries, indicating the sample size is sufficient to get significant results.

## Sensitivity analysis

Sensitivity analysis was performed to analyze the impact of each study on the net results. In our study, the pooling effect was no significant difference before and after excluding each study with high heterogeneity one by one, indicating the results of sensitivity analysis are stable (S3 Fig).

## Publication bias

As is shown in S2 Table, the results of the Egger's test for the analysis of EPS-related risk factors among PD patients showed that the included studies may have no possibility of publication bias (P>0.05).

## Discussion

EPS, known for its high mortality, is one of the most serious complications of PD. Although many studies have reported risk factors for the occurrence of EPS, the data remains controversial. To the best of our knowledge, this is the first meta-analysis summarizing the risk factors of EPS. To accurately determine the risk factors related to the occurrence of EPS, we collected the relevant data from ten observational studies in eight countries or regions. Differences are found in the incidence of EPS reported in various reports (from 0.4% to 11.7%). It is worth noting that 2 important factors may influence this result. First, the eight studies included in this study are retrospective study designs, which may raise the possibility of recall bias. Second, the use of PD fluid in the studies differed slightly. For example, conventional PD fluid, biocompatible PD fluid and icodextrin were used in the report of Nakao et al, instead, all the patients used standard Dianeal dialysis solutions in the report of Alatab et al. Finally, lower age at PD onset, higher D/P Cr, longer duration of PD, longer peritonitis duration, and presence of GN were identified as significant risk factors for EPS.

This study demonstrated that age was a risk factor for the development of EPS, wherein the age of patients with EPS was lower than that of the non-EPS group (MD = -7.70, 95% CI, -11.53~-3.86, P < 0.0001). Similarly, a retrospective study showed that the average age at PD onset of 12 patients with EPS (35.1 ± 3.3 years) was significantly lower than that of patients on PD without EPS (47.3 ± 1.1 years), and the difference was as much as 12 years [30]. Given its independent importance, this variable is not only a consequence of longer duration on PD or longer follow-up after PD onset. Mesothelial cells play a key role in the process of peritoneal remodelling [32]. Mesothelial to mesenchymal transition is a main mechanism of peritoneal fibrosis in which mesothelial cells secrete cytokines during exposure to multiple insults (e.g. glucose) that subsequently result in the recruitment of fibroblasts and macrophages [20, 33]. Perhaps this disrupted peritoneal remodeling function is more vigorous in younger patients, which might be the main reason for the earlier development of fibrosis.

D/P Cr is an international index for evaluating the state of peritoneal transport. In the studies reported by Yamamoto et al. [19, 30], they defined a D/P Cr greater than 0.75 as a higher peritoneal transport state, which was found to be an important independent risk factor for EFS. Therefore, we sought to elucidate whether an increased peritoneal D/P Cr ratio might serve as an indicator of the early development of EPS. This study revealed that the value of D/P Cr was higher in the EPS group than in the non-EPS patient group, which indicates that the peritoneal transport status affects the incidence of EPS [34–36]. The development of a high peritoneal transport state is multi-factorial, partly because of the changes in the morphology of the capillaries supplying the peritoneum, and the effects of some humoral factors, such as angiogenic factors, cannot also be ruled out [37, 38]. These changes would ultimately trigger peritoneal sclerosis. More research, therefore, is necessary to focus on the relationship between high transport state and EPS in the future.

The results of our meta-analysis showed that the duration of PD was positively correlated with the incidence of EPS (SMD = 1.15, 95% CI, 0.68–1.61, P < 0.001). PD duration has been regarded as the most important risk factor for EPS development in several studies [3, 39–41]. For example, in a retrospective study, the overall incidence of EPS was only 0.3% (1/353) within the first four years of PD, whereas this figure reached 4.2% (3/71), 41.6% (5/12) and 27.3% (3/11) for patients with PD therapy for 4–6, 6–8 and >8years, respectively [21]. Similarly increase in EPS incidence also reported by other studies [12–14, 16, 27, 40–42]. Long-term PD can affect the peritoneal function and increase the number of peritonitis episodes, which might partly account for this phenomenon. For high-risk patients, discontinuing PD after 8 years might be an effective measure to prevent the occurrence of EPS [8].

Although peritonitis episodes particularly caused by fungal or bacterial organisms have long been regarded as a significant risk factor for EPS development [14, 30, 43–45], contradictory studies also exist. A Japanese prospective study reported that 25% of patients who developed EPS were related bacterial peritonitis [13], whereas other studies could not showed an associated between peritonitis episodes and EPS [4, 12, 27, 46–48]. This meta-analysis also indicated that the number of peritonitis episodes might not be relevant to EPS development. The ISPD, however, suggested that patients on PD with severe and/or non-resolving peritonitis have a higher tendency to develop EPS [2]. Yamamote and colleagues explained this result, suggesting that the occurrence of peritonitis-induced EPS might be affected by the duration of PD, thereby exaggerating the risk assessment between the two [30]. This conclusion needs to be further confirmed in future studies. Interestingly, previous studies have emphasized that it is the duration of peritonitis that increases the risk of EPS [21, 22, 28–30, 49]. This association can provide valuable information for the early detection and treatment of EPS.

Our study also found that the risk of EPS occurrence was 1.42 times higher in GN patients than in PD patients without GN. As many articles have reported, Tacrolimus and cyclosporine can promote fibrosis progression possibly by activating the epithelial-mesenchymal transition(EMT) response (increased transforming growth factor β1 and α smooth muscle actin expression) and increasing the expression of fibrose-related genes [50–52]. Nevertheless, there was no information about previous drugs used in the glomerulonephritis group. This should be one of the present study limitations. Hence, before drawing definitive conclusions, additional research is necessary.

The advantages of this study are adapting comprehensive search, including large sample size studies and assessing the specific risk factors using the meta-analytic method. Nevertheless, several limitations of this study must be considered when interpreting the results. First, we included a total of 10 papers in our meta-analysis, of which 4 papers originated from Japan; this could be a reason for bias. Second, medication history was not provided in these literatures, which might affect outcomes of patients. Finally, the validity of statistical analyses was affected by the disparities in heterogeneity among studies. Due to the limited number of studies, we only performed sensitivity analysis without subgroup analysis. Nevertheless, our results made an important contribution to the identification of well-recognized risk factors associated with EPS by integrating studies involving risk factors for EPS.

## Conclusions

In conclusion, EPS is a rare and most serious complication in patients on long-term PD. This study comprehensively summarized EPS-related risk factors and concluded that lower age at PD onset, higher D/P Cr, longer duration of PD, longer peritonitis duration, and history of GN are associated with significantly increased risk of EPS among PD patients. These findings provide a scientific basis for further understanding the etiology of PD-related EPS and improving prevention strategies. Particularly, the incidence of EPS can be reduced by controlling a series of modifiable risk factors. More high-quality studies are needed to validate this paper's findings in the future.

## Supporting information

**S1 Fig. TSA results of the related risk factors.** (A) Age at PD onset; (B) D/P Cr. The information size was calculated based on MD, alpha of 5%, power of 80%.
(TIF)

**S2 Fig. TSA results of the related risk factors.** (A) Peritonitis duration; (B) History of GN. The information size was calculated based on relative risk reduction (RRR) or MD, alpha of

5%, power of 80%.
(TIF)

**S3 Fig. Results of sensitivity analysis.** (A) Age on dialysis. (B) D/P Cr. (C) PD duration. (D) peritonitis duration. (E) Use of icodextrin.
(TIF)

**S1 File. Checklist.** PRISMA 2020 checklist.
(DOCX)

**S2 File. Search strategy for the databases.**
(DOCX)

**S3 File. The protocol of this review.**
(PDF)

**S1 Table. Risk assessment of the included studies using ROBINS-I tool.**
(DOCX)

**S2 Table. Publication bias associated with potential risk factors for EPS.**
(DOCX)

## Acknowledgments

We would like to express our gratitude to all who contributed to the writing of the reviewed articles in this meta-analysis.

## Author Contributions

**Conceptualization:** Dashan Li, Yonggui Wu.

**Data curation:** Dashan Li, Yuanyuan Li, Hanxu Zeng, Yonggui Wu.

**Formal analysis:** Dashan Li, Yonggui Wu.

**Methodology:** Hanxu Zeng.

**Writing – original draft:** Dashan Li.

**Writing – review & editing:** Dashan Li, Hanxu Zeng, Yonggui Wu.

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
