## [Decision Letter · Decision Letter 0]

5 Jan 2022

PONE-D-21-39037Risk factors for Encapsulating Peritoneal Sclerosis in patients undergoing peritoneal dialysis: a meta-analysisPLOS ONE

Dear Dr. Yonggui Wu,

Thank you for submitting your manuscript to PLOS ONE. After careful consideration, we feel that it has merit but does not fully meet PLOS ONE’s publication criteria as it currently stands. Therefore, we invite you to submit a revised version of the manuscript that addresses the points raised during the review process.

We look forward to receiving your revised manuscript.

Kind regards,

Wisit Cheungpasitporn, MD

Academic Editor

PLOS ONE

Journal Requirements:

Additional Editor Comments:

The reviewers have raised a number of points which we believe major modifications are necessary to improve the manuscript, taking into account the reviewers' remarks. Please consider and address each of the comments raised by the reviewers before resubmitting the manuscript. This letter should not be construed as implying acceptance, as a revised version will be subject to re-review.

Reviewers' comments:

Reviewer's Responses to Questions

**Comments to the Author**

1. Is the manuscript technically sound, and do the data support the conclusions?

Reviewer #1: Yes

Reviewer #2: No

Reviewer #3: Yes

2. Has the statistical analysis been performed appropriately and rigorously? 

Reviewer #1: I Don't Know

Reviewer #2: No

Reviewer #3: Yes

3. Have the authors made all data underlying the findings in their manuscript fully available?

Reviewer #1: Yes

Reviewer #2: Yes

Reviewer #3: Yes

4. Is the manuscript presented in an intelligible fashion and written in standard English?

Reviewer #1: Yes

Reviewer #2: No

Reviewer #3: Yes

5. Review Comments to the Author

Reviewer #1: 1. In the introduction (Line 97), risk factors for EPS have been mentioned which includes duration of PD, age at initiation, bio incompatible dialysate, number of peritonitis and duration, kidney transplantation.

Comment: Organ transplantation has been mentioned to increase the risk of EPS possibly related to use of immunosuppressive medications or withdrawal of PD (Fieren et al, Posttransplant encapsulating peritoneal sclerosis: a worrying new trend?).

In your meta-analysis, age at PD onset was significantly associated with development of EPS.

Given the number of young PD patients who undergo transplantation or have access to transplantation, I wonder if younger age at onset of PD and transplantation puts them at a higher risk of developing EPS in future. If data is available in the included case control studies, I would be curious to know if transplantation increased risk of EPS in your meta-analysis and consider addition of that to your manuscript.

2. Although young age has been implicated as a risk factor in your study and multiple studies before, is it confounded by the fact that younger patients may undergo PD for a longer duration of time, undergo transplantation and are alive long enough to develop EPS while older patients may not undergo PD for a longer duration and may not be alive long enough to develop EPS?

3. Use of Icodextrin was not a risk factor for development of EPS. I wonder if using Dianeal 2.5/4.25% is associated with higher risk of development of EPS as icodextrin may not be used for an entire dwell and since icodextrin is a water-soluble polymer may behave differently when compared to dextrose. Was there any information on other types of PD fluids used in the included studies?

Overall, I commend the authors for this well conducted meta-analysis to determine the risk factors associated with EPS.

Reviewer #2: I thank the authors and editors for this opportunity to review this manuscript.

My comments are as below:

1. The main weakness of the study lies in its eligibility criteria. We know that the aggregated evidence from observational studies is weaker than randomized controlled trials. In this regard, restricting the inclusion criteria to a particular type of observational study compromises the robustness and comprehensiveness of the review. Next, the study population remains poorly described. Moreover, the exclusion criteria didn't appear sound to me. The statement (sentence no. 130-132) of not recruiting poor-quality studies is more of a subjective decision of the authors and is not an endorsable way to eliminate studies.

2. The meta-analysis model choice guided by heterogeneity assessment is not recommended.

3. The cut-offs used to determine and quantify heterogeneity are not clear.

4. It's not advisable to use funnel plots for publication bias assessment when there are <10 studies available for inclusion in the meta-analysis models.

5. The abstract is not clear enough. I suggest a more simplified and logical presentation.

6. The total capitalization of some of the database names (e.g., PubMed) is not advisable.

7. The phrase "qualified studies" is better avoided.

8. Sentence 31-32 in the abstract is confusing. Ideally, co-stating the rationale of mentioning alternative model choices is required.

9. Sentence 44-45 in the abstract doesn't fit a conclusion statement. Better to avoid such in the abstract.

10. The study lacks a protocol to compare its pre-review notions to the presented methodology.

11. I couldn't find any mention of the supplementary tables and figures in the main manuscript text. These are simply listed before the reference section.

12. It's not clear how PubMed search strings got combined.

13. The claim of searching all relevant data as mentioned in sentence no. 116-117 is perhaps not appropriate.

14. The type of conflicting data authors is referring to in sentence no. 121-122 is not clear.

15. The tables lack publication-quality legends, footers, and formatting.

16. Sentence no. 150-152 is confusing.

17. Regarding sentence no. 157-159, without a study protocol, it's impossible to compare it with the preliminary plan made by the authors.

18. The PRISMA flow diagram is not adequately cited and presented.

19. The PRISMA checklist is not recommended anymore as a more recent version is available.

20. The figures lack proper legends. The requirements of different colors in forest plots are unclear.

21. The manuscript writing requires meticulous editing.

22. References use was inadequate.

Thank you.

Reviewer #3: Manuscript ID: PONE-D-21-39037

The authors present a comprehensive summary of their systematic review and meta-analysis of Risk factors for Encapsulating Peritoneal Sclerosis in patients undergoing peritoneal dialysis: a meta-analysis. They presented that lower age at PD onset, higher D/P Cr, longer PD duration, longer duration of peritonitis, and history of glomerulonephritis were risk factors for EPS. This review is interesting. However, there are some concerns that need to be addressed.

1. Grey literature: If the authors performed the search using specific gray literature, please provide the name of such database. Overall, I have concerns about the reproducibility of the meta-analysis. Certain aspects of the MOOSE guideline were followed, but there are some key flaws that may affect the summary effect estimate. Since search limited, unpublished studies were not included, I worry that this summary effect estimate will shrink or be nullified if grey literature, unpublished studies are found. The authors may only be able to say that their meta-analysis resolves discrepancies between published studies. I am hesitant because there are too many examples of unpublished data nullifying significant effects in meta-analyses. If the authors conduct a thorough search for unpublished data, this comment becomes irrelevant. Please add this information in supplementary.

2. This review was not followed PRISMA 2020 flow and absent of PRISMA 2020 checklist. The author should follow PRISMA 2020 flow diagram structure. In addition, the protocol registration was absent. Prospective registration of systematic reviews promotes transparency, helps reduce potential for bias and serves to avoid unintended duplication of reviews.

3. In "Methods" part, the independent investigators who extracted data and evaluated quality of each study should be clarify.

4. How were the search terms defined? Is there a pre-test to define the search strategy used in each database? Did the search strategy the same in all databases? I wonder if the search strategies have been developed with the help of a librarian or experienced reviewers in the field. Generally, if the search yield is too low, systematic reviewers would need to modify the search term to ensure that it well cover most of the related papers. This could be done by in the PICO such as Control and Outcome; employ a free text rather than thesaurus search terms. I think the keywords for the outcome domain were not comprehensive enough to capture all potential synonyms of the outcomes of interested in this study, and therefore would be best not to limit the search with these terms. An example of the search strategies used for a particular database would show a transparency in this step (I don’t think the keywords presented is sufficient as an example of a search strategies as recommended in PRISMA).

5. The date of the literature was April 1, 2021. In accordance with guidelines, the literature search should be performed until six months before the submission of the manuscript for publication. If the manuscript is accepted for publication, it is already outdated. In addition, the author should describe exact date and duration of the literature search in both abstract and main manuscript.

6. How authors dealt with missing data. Did you receive all answers from authors of the studies or make some imputation? In general, if the study did not report the data of the primary or secondary outcomes measures, the authors should contact via email to provide this information. Have you considered to contact authors via researchers’ network ResearchGate (https://www.researchgate.net/), Academia (https://www.academia.edu/), Loop (https://loop.frontiersin.org/) or Quora (https://es.quora.com/)? Nowadays these platforms are very useful and efficient canals to contact authors.

7. In quality assessment for observational study design part, I recommend the authors apply the ROBINS-I (Risk of Bias in Nonrandomized studies of Interventions) tool. The authors already applied the Newcastle Ottawa Scale, which is a validated tool and was an acceptable choice. However, to enhance the reproducibility and comparability of this review to future reviews of a similar topic (possibly an update of this review) I recommend including a risk of bias assessment using ROBINS-I, since it is the newest and most robust method of assessing risk of bias in systematic reviews/meta-analyses.

8. The authors should demonstrate both statistics and visualization. In addition, I suggest plot the funnel and contour-enhanced funnel in the graphic and not only the studies for better interpretation. Besides, it is necessary to present the p value for this analysis.

9. Finally, since I am not a native English user, I did not check for grammatical errors thoroughly. This should be done by an appropriate language reviewer.

6. PLOS authors have the option to publish the peer review history of their article (what does this mean?). If published, this will include your full peer review and any attached files.

Reviewer #1: No

Reviewer #2: **Yes: **Sumanta Saha

Reviewer #3: **Yes: **Wisit Kaewput

---

## [Author Response · Author response to Decision Letter 0]

20 Feb 2022

Response to comment

We appreciate the time and efforts by the editor and referees in reviewing this manuscript. We have addressed all issues indicated in the review report, which may largely improve our current study.

Dear Dr. Yonggui Wu,

Thank you for submitting your manuscript to PLOS ONE. After careful consideration, we feel that it has merit but does not fully meet PLOS ONE’s publication criteria as it currently stands. Therefore, we invite you to submit a revised version of the manuscript that addresses the points raised during the review process.

If applicable, we recommend that you deposit your laboratory protocols in protocols.io to enhance the reproducibility of your results. Protocols.io assigns your protocol its own identifier (DOI) so that it can be cited independently in the future. For instructions see:https://journals.plos.org/plosone/s/submission-guidelines#loc-laboratory-protocols. Additionally, PLOS ONE offers an option for publishing peer-reviewed Lab Protocol articles, which describe protocols hosted on protocols.io. Read more information on sharing protocols athttps://plos.org/protocols?utm_medium=editorial-email&utm_source=authorletters&utm_campaign=protocols.

We look forward to receiving your revised manuscript.

Kind regards,

Wisit Cheungpasitporn, MD

Academic Editor

PLOS ONE

Answer: Thanks very much for your positive comments and kind reminding. We have revised the manuscript as suggested and re-submit our manuscript for fresh review. Meanwhile, This meta-analysis was registered with the International Prospective Register of Systematic Reviews (PROSPERO) (CRD42022302786). The protocol of this review was presented in supplementary S3 File. Thank you again.

Journal Requirements:

2. https://journals.plos.org/plosone/s/file?id=wjVg/PLOSOne_formatting_sample_main_body.pdf and https://journals.plos.org/plosone/s/file?id=ba62/PLOSOne_formatting sample_title_authors_affiliations.pdf

Answer: Thank you for your time in reviewing this manuscript. We have revised the format of this manuscript again based on PLOS ONE's style requirements, including file naming, tables formatting, and references formatting. Thank you again.

Answer: Thanks very much for your suggestion. We have updated the ORCID iD for the corresponding author.

Reviewers' Comments to Author:

Reviewer: 1

Comments to the Author

Major comments

1. Organ transplantation has been mentioned to increase the risk of EPS possibly related to use of immunosuppressive medications or withdrawal of PD (Fieren et al, Posttransplant encapsulating peritoneal sclerosis: a worrying new trend?). In your meta-analysis, age at PD onset was significantly associated with development of EPS. Given the number of young PD patients who undergo transplantation or have access to transplantation, I wonder if younger age at onset of PD and transplantation puts them at a higher risk of developing EPS in future. If data is available in the included case control studies, I would be curious to know if transplantation increased risk of EPS in your meta-analysis and consider addition of that to your manuscript.

Answer: Thank you for arising this question. In some studies, a high incidence of post-transplantation EPS has been noted [ref. 1. Balasubramaniam, G, Nephrol Dial Transplant. 2009(PMID：19211652); 2. Afthentopoulos, I, Adv Ren Replace Ther. 1998(PMID：9686626)]. Balasubramaniam G, et al reported that the median time from renal transplantation to development of EPS was 5.4 months. As you said, The use of calcineurin inhibitor as a part of immunosuppressive regimens also increased the risk of EPS. For example, Tacrolimus and cyclosporine can promote fibrosis progression possibly by activating the epithelial-mesenchymal transition(EMT) response (increased TGF-β1 and α-SMA expression) and increasing the expression of fibrose-related genes [ref. 1. Liu QF, Int Urol Nephrol. 2017(PMID: 27796696); 2. Nam HK, Am J Nephrol. 2013(PMID: 23258196); 3. Lim BJ, Pediatr Nephrol. 2009(PMID：19066978); 4. Kern G, PLoS One. 2014(PMID：24816588)]. 

However, only one study reported some Variables related to kidney transplantation [Korte MR, Perit Dial Int. 2011(PMID：21454391)] between EPS group and NO EPS group, which demonstrated that a typical patient with EPS after kidney transplantation has started on PD at relatively young age (<50 years). Therefore, the meta-analysis for kidney transplantation was not performed here. Thank you again.

2. Although young age has been implicated as a risk factor in your study and multiple studies before, is it confounded by the fact that younger patients may undergo PD for a longer duration of time, undergo transplantation and are alive long enough to develop EPS while older patients may not undergo PD for a longer duration and may not be alive long enough to develop EPS?

Answer: Thank you for arising this question. Several studies have confirmed that the younger a patient is when at PD start, the greater the chance that EPS will develop. [ref. 1. Yamamoto R, Adv Perit Dial. 2002(PMID：12402604); 2. Johnson DW, Kidney Int. 2010(PMID：20375981)]. Given its in dependent significance, this variable is not simply a result of a longer time on PD or a longer duration of follow-up from PD initiation. As many articles have reported, they proposed the disrupted peritoneal fibrosis repair process in PD patients with younger age as possible cause of this observation [ref. 1. Korte MR, Perit Dial Int. 2011(PMID：21454391); 2. Chin AI, Am J Kidney Dis. 2006(PMID：16564950)].

In reaction to multiple insults with (for example) glucose, mesothelial cells secrete cytokines that subsequently lead to recruitment of macrophages and fibroblasts. Peritoneal fibrosis is then a result of disrupted repair, with fibrin deposition on a denudated mesothelial cell layer. Perhaps this repair process is more vigorous in younger patients, leading to earlier development of fibrosis. Therefore, we considered the age at PD onset as a risk factor for the development of EPS. We have added more explanations for this variable into the manuscript (Line 412-421, Page 23-24). Thank you again.

3. Use of Icodextrin was not a risk factor for development of EPS. I wonder if using Dianeal 2.5/4.25% is associated with higher risk of development of EPS as icodextrin may not be used for an entire dwell and since icodextrin is a water-soluble polymer may behave differently when compared to dextrose. Was there any information on other types of PD fluids used in the included studies?

Answer: Thank you for arising this question. As many articles have reported, PD fluids contain glucose, which is a major cause of peritoneal membrane injury. Heat sterilization of PD fluids also leads to the formation of glucose degradation products, which accelerate the formation of AGEs [ref. Moriishi, M, Adv Perit Dial. 2002(PMID：12402608)]. Moreover, patients who developed EPS had a higher cumulative glucose exposure [ref. 1. Lambie, M. L, Kidney Int. 2010(PMID：20571473); 2. Hendriks, P. M, Perit Dial Int. 1997(PMID：9159833)].

However, there was no reported data about other type of PD fluids used between EPS group and NO EPS group among studies. Thank you again.

Reviewer: 2

Comments to the Author

 I thank the authors and editors for this opportunity to review this manuscript.

Answer:Thanks very much for your comments. We revised the manuscript as suggested.

Major comments

1. The main weakness of the study lies in its eligibility criteria. We know that the aggregated evidence from observational studies is weaker than randomized controlled trials. In this regard, restricting the inclusion criteria to a particular type of observational study compromises the robustness and comprehensiveness of the review. Next, the study population remains poorly described. Moreover, the exclusion criteria didn't appear sound to me. The statement (sentence no. 130-132) of not recruiting poor-quality studies is more of a subjective decision of the authors and is not an endorsable way to eliminate studies.

Answer: Thanks very much for your suggestion. We sincerely appreciate the valuable comment. Firstly, we re-searched relevant citations in five databases and one website from their inception to January 1st, 2022 without restriction by study type, and two prospective cohort studies included in this meta-analysis. Correspondingly, the contents of tables and figures in this study have been modified. Secondly, We have modified the content in Table 1, in which the basic characteristics of study population were further described (Page 12). Finally, We've revised our exclusion criteria in this study (Line 140-142, Page 7).

2. The meta-analysis model choice guided by heterogeneity assessment is not recommended.

Answer: Thanks very much for your suggestion. In PRISMA 2020, the fixed-effect model assumes that there is a common treatment effect for all included studies; it is assumed that the observed differences in results across studies reflect random variation. The random-effects model assumes that there is no common treatment effect for all included studies but rather that the variation of the effects across studies follows a particular distribution. However, differences in study design, interventions, and study populations of the included studies make it difficult to decide on the choice of meta-analysis model. Therefore, we intuitively selected a model based on the heterogeneity statistics observed. Thank you again.

3. The cut-offs used to determine and quantify heterogeneity are not clear.

Answer: Thank you for arising this question. In this meta-analysis, when I2 was > 50% and Q chisquared test result < 0.1, it shows that there is large heterogeneity among the trials. Once large heterogeneity was present, sensitivity analysis were performed to identify responsible outlier studies. Therefore, we deleted the Fig 5 and the results of sensitivity analysis were presented in S3 Fig. We have revised the content in the part of statistical analysis with red markers (Line 165-169, Page 9). Thank you again.

4. It's not advisable to use funnel plots for publication bias assessment when there are <10 studies available for inclusion in the meta-analysis models.

Answer: Thanks very much for your suggestion. Publication bias was evaluated using the Egger’s test in this study (Line 170, Page 9). The results of the Egger’s test for the analysis of EPS-related risk factors among PD patients were presented in S2 Table and showed that the included studies may have no possibility of publication bias. Thank you again.

5. The abstract is not clear enough. I suggest a more simplified and logical presentation.

Answer: Thanks very much for your suggestion. We removed unnecessary information, added the description of risk assessment and revised some data with red markers to make the abstract more simplified.

6. The total capitalization of some of the database names (e.g., PubMed) is not advisable.

Answer: Thanks very much for pointing out my mistakes. We have revised this error into the correct statement (Line 28, Page 2) (Line 120, page 6). 

7. The phrase "qualified studies" is better avoided.

Answer: Thanks very much for your suggestion. We have revised this statement with red markers (Line 122-123, Page 6-7).

8. Sentence 31-32 in the abstract is confusing. Ideally, co-stating the rationale of mentioning alternative model choices is required.

Answer: Thanks very much for your suggestion. To further simplify the presentation of the abstract, we removed the sentence 31-32 and added the description of risk assessment in the part of methods with red markers (Line 31-33, Page 2).

9. Sentence 44-45 in the abstract doesn't fit a conclusion statement. Better to avoid such in the abstract.

Answer: Thanks very much for your suggestion. We removed sentence 44-45 and re-written the conclusion section of the abstract. Thank you again.

10. The study lacks a protocol to compare its pre-review notions to the presented methodology.

Answer: Thanks very much for your suggestion. As you suggested, this meta-analysis was registered at PROSPERO (CRD42022302786). The protocol of this review was presented in S3 File.

11. I couldn't find any mention of the supplementary tables and figures in the main manuscript text. These are simply listed before the reference section.

Answer: Thank you for arising this question. the supplementary tables and figures captions were listed after the reference section with red markers. Meanwhile, the contents of supporting information files were revised and uploaded separately in submission system. Thank you again.

12. It's not clear how PubMed search strings got combined.

Answer: Thank you for arising this question. We used a combination of search medical subject heading (MeSH) related to EPS and risk factor in PD patients. We made some changes to the search terms and the final strategies were peer reviewed by an experienced information specialist within our team. S2 File outlined the detailed search strategy of PubMed. Thank you again.

13. The claim of searching all relevant data as mentioned in sentence no. 116-117 is perhaps not appropriate.

Answer: Thanks very much for your suggestion. We have revised this statement with red markers(Line 120, Page 6).

14. The type of conflicting data authors is referring to in sentence no. 121-122 is not clear.

Answer: Thank you for arising this question. Conflicting data in sentence no. 121-122 refers to any discrepancies in study selection processes. We have revised this statement with red markers (Line 127, Page 7).

15. The tables lack publication-quality legends, footers, and formatting.

Answer: Thanks very much for your suggestion. We have revised format of the tables and made the necessary changes to the content of the tables. Legends and footnotes have been placed below the tables. Thank you again.

16. Sentence no. 150-152 is confusing. 

Answer: Thank you for arising this question. In this study, continuous variables were summarized using mean difference (MD) or standardized mean difference (SMD) with their 95% confidence intervals (CI), for dichotomous variables, odds ratio (OR) with their 95% CI were estimated. We revised the sentence no. 150-152 in the section of statistical analysis with red markers (Line 161-164, Page 8).

17. Regarding sentence no. 157-159, without a study protocol, it's impossible to compare it with the preliminary plan made by the authors.

Answer: Thank you for arising this question. This is a typo. Because large heterogeneity was found in potential risk factors of exposure to age at PD onset, duration of PD, D/P Cr, peritonitis duration, and use of icodextrin, we wanted to perform subgroup analysis to explore sources of heterogeneity. However, we only performed sensitivity analysis due to the limited number of studies and data. We have deleted this inappropriate statement. Thank you again.

18. The PRISMA flow diagram is not adequately cited and presented.

Answer: Thanks very much for your suggestion. As you suggested, we have followed PRISMA 2020 flow diagram structure (Fig.1). 

19. The PRISMA checklist is not recommended anymore as a more recent version is available.

Answer: Thanks very much for your suggestion. The PRISMA 2020 checklist has already uploaded again (S1 File). Thank you again.

20. The figures lack proper legends. The requirements of different colors in forest plots are unclear.

Answer: Thank you for arising this question. We added the proper descriptions in figures legends with red markers. In addition, different colors in forest plots represent different types of primitive data (e.g. green refers to Continuous variables, blue refers to dichotomous variables). The colors in these forest plots cannot be changed in the analysis software. Thank you again.

21. The manuscript writing requires meticulous editing.

Answer: Thanks very much for your suggestion. We have deleted some inappropriate presentations and polished some sentences as well the corresponding revision into manuscript with red markers in the sections of abstract, methods, results, discussion, and conclusion. Meanwhile, We invited native speakers to help us revise the grammar errors. Thank you again.

22. References use was inadequate.

Answer: Thanks very much for your suggestion. We have added more references for further verification in this study. Thank you again.

Reviewer: 3

Comments to the Author

The authors present a comprehensive summary of their systematic review and meta-analysis of Risk factors for Encapsulating Peritoneal Sclerosis in patients undergoing peritoneal dialysis: a meta-analysis. They presented that lower age at PD onset, higher D/P Cr, longer PD duration, longer duration of peritonitis, and history of glomerulonephritis were risk factors for EPS. This review is interesting. However, there are some concerns that need to be addressed.

Answer:Thanks very much for your comments. We revised the manuscript as suggested.

Major comments

1. Grey literature: If the authors performed the search using specific gray literature, please provide the name of such database. Overall, I have concerns about the reproducibility of the meta-analysis. Certain aspects of the MOOSE guideline were followed, but there are some key flaws that may affect the summary effect estimate. Since search limited, unpublished studies were not included, I worry that this summary effect estimate will shrink or be nullified if grey literature, unpublished studies are found. The authors may only be able to say that their meta-analysis resolves discrepancies between published studies. I am hesitant because there are too many examples of unpublished data nullifying significant effects in meta-analyses. If the authors conduct a thorough search for unpublished data, this comment becomes irrelevant. Please add this information in supplementary.

Answer: Thanks very much for your suggestion. We sincerely appreciate the valuable comment. We searched the http://www.greylit.org/ website for studies that were registered as completed but not yet published. However, no relevant studies were registered in this website. We have added it to the research strategy with red markers (Line 123-125, Page 7). Thank you again.

2. This review was not followed PRISMA 2020 flow and absent of PRISMA 2020 checklist. The author should follow PRISMA 2020 flow diagram structure. In addition, the protocol registration was absent. Prospective registration of systematic reviews promotes transparency, helps reduce potential for bias and serves to avoid unintended duplication of reviews.

Answer: Thanks very much for your suggestion. As you suggested, we have followed PRISMA 2020 flow diagram structure (Fig 1) and the PRISMA 2020 checklist has already uploaded again (S1 File). In addition, this meta-analysis was registered at PROSPERO (CRD42022302786). The protocol of this review was presented in S3 File. Thank you again.

3. In "Methods" part, the independent investigators who extracted data and evaluated quality of each study should be clarify.

Answer: Thanks very much for your suggestion. In this study, data extraction and quality assessment were completed by two authors (Yuanyuan Li, and Hanxu Zeng), which were detailed in the methods section (Line 129, Page 7) (Line 145, Line 150 Line 157, Page 8).

4. How were the search terms defined? Is there a pre-test to define the search strategy used in each database? Did the search strategy the same in all databases? I wonder if the search strategies have been developed with the help of a librarian or experienced reviewers in the field. Generally, if the search yield is too low, systematic reviewers would need to modify the search term to ensure that it well cover most of the related papers. This could be done by in the PICO such as Control and Outcome; employ a free text rather than thesaurus search terms. I think the keywords for the outcome domain were not comprehensive enough to capture all potential synonyms of the outcomes of interested in this study, and therefore would be best not to limit the search with these terms. An example of the search strategies used for a particular database would show a transparency in this step (I don’t think the keywords presented is sufficient as an example of a search strategies as recommended in PRISMA).

Answer: Thank you for arising this question. We attached great importance to this comment. As you proposed, we redefined the search terms based on a PICO-style approach with the help of a reviewer. Firstly, five known relevant studies were used to identify records within databases. Secondly, Candidate search terms were identified by looking at words in the titles, abstracts and subject indexing of those records. A draft search strategy was developed using those terms and additional search terms were identified from the results of that strategy. Search terms were also identified and checked using the PubMed PubReMiner word frequency analysis tool. Finally, The strategy was developed by an information specialist and the final strategies were peer reviewed by an experienced information specialist within our team. The Search terms in each database were the same, while search strategy was different. S2 File outlined the detailed search strategy of PubMed. Thank you again.

5. The date of the literature was April 1, 2021. In accordance with guidelines, the literature search should be performed until six months before the submission of the manuscript for publication. If the manuscript is accepted for publication, it is already outdated. In addition, the author should describe exact date and duration of the literature search in both abstract and main manuscript.

Answer: Thanks very much for your valuable suggestion. We re-searched relevant citations in five databases and one website from their inception to January 1st, 2022, which were detailed in the sections of abstract and methods and made some changes to the search terms. A total of 495 potentially relevant articles were retrieved initially. Finally, we included 10 standards-compliant articles. Correspondingly, the contents of tables and figures in this study have been modified. Thank you again.

6. How authors dealt with missing data. Did you receive all answers from authors of the studies or make some imputation? In general, if the study did not report the data of the primary or secondary outcomes measures, the authors should contact via email to provide this information. Have you considered to contact authors via researchers’ network ResearchGate (https://www.researchgate.net/), Academia (https://www.academia.edu/), Loop (https://loop.frontiersin.org/) or Quora (https://es.quora.com/)? Nowadays these platforms are very useful and efficient canals to contact authors.

Answer: Thank you for arising this question. We were lucky that in this study, there was no data missing in the extraction process.

7. In quality assessment for observational study design part, I recommend the authors apply the ROBINS-I (Risk of Bias in Nonrandomized studies of Interventions) tool. The authors already applied the Newcastle Ottawa Scale, which is a validated tool and was an acceptable choice. However, to enhance the reproducibility and comparability of this review to future reviews of a similar topic (possibly an update of this review) I recommend including a risk of bias assessment using ROBINS-I, since it is the newest and most robust method of assessing risk of bias in systematic reviews/meta-analyses.

Answer: Thanks very much for your valuable suggestion. As you suggested, potential bias was evaluated by the ROBINS-I tool for observational studies. Finally, all studies presented a moderate overall risk of bias, in which confounding bias remains in all studies due to their non-randomized nature. The exact details of the risk evaluation are summarized in S1 Table. Thank you again.

8. The authors should demonstrate both statistics and visualization. In addition, I suggest plot the funnel and contour-enhanced funnel in the graphic and not only the studies for better interpretation. Besides, it is necessary to present the p value for this analysis.

Answer: Thanks very much for your suggestion. However, one reviewer suggested that the use of funnel plots for publication bias assessment is inappropriate in this study when there are ≤10 studies available for inclusion in the meta-analysis models. Therefore, publication bias of included studies was evaluated again using the Egger’s test. The results of Egger’s test were presented in S2 Table. Thank you again.

9. Finally, since I am not a native English user, I did not check for grammatical errors thoroughly. This should be done by an appropriate language reviewer.

 Answer: Thanks for your time in reviewing this manuscript. We have invited native speakers to help us revise the grammar errors. 

We tried our best to improve the manuscript and made some changes in the manuscript. These changes will not influence the content and framework of the paper. And here we did not list the changes but marked in red in revised paper.

We appreciate for Editors/Reviewers’ warm work earnestly, and hope that the correction will meet with approval. Once again, thank you very much for your comments and suggestions.

---

## [Decision Letter · Decision Letter 1]

7 Mar 2022

Risk factors for Encapsulating Peritoneal Sclerosis in patients undergoing peritoneal dialysis: a meta-analysis

PONE-D-21-39037R1

Dear Dr. wu,

We’re pleased to inform you that your manuscript has been judged scientifically suitable for publication and will be formally accepted for publication once it meets all outstanding technical requirements.

Kind regards,

Wisit Cheungpasitporn, MD

Academic Editor

PLOS ONE

Additional Editor Comments:

It appears that all comments have been appropriately responded to. I have no further comments and recommend publication.

Reviewers' comments:

Reviewer's Responses to Questions

**Comments to the Author**

1. If the authors have adequately addressed your comments raised in a previous round of review and you feel that this manuscript is now acceptable for publication, you may indicate that here to bypass the “Comments to the Author” section, enter your conflict of interest statement in the “Confidential to Editor” section, and submit your "Accept" recommendation.

Reviewer #1: All comments have been addressed

Reviewer #3: All comments have been addressed

2. Is the manuscript technically sound, and do the data support the conclusions?

Reviewer #1: Yes

Reviewer #3: Yes

3. Has the statistical analysis been performed appropriately and rigorously? 

Reviewer #1: I Don't Know

Reviewer #3: Yes

4. Have the authors made all data underlying the findings in their manuscript fully available?

Reviewer #1: Yes

Reviewer #3: Yes

5. Is the manuscript presented in an intelligible fashion and written in standard English?

Reviewer #1: Yes

Reviewer #3: Yes

6. Review Comments to the Author

Reviewer #1: No further comments to the authors. Authors have addressed all my comments. Manuscript is well written.

Reviewer #3: Thank you. I am satisfied with the authors replies and modifications and have no further suggestions.

7. PLOS authors have the option to publish the peer review history of their article (what does this mean?). If published, this will include your full peer review and any attached files.

Reviewer #1: No

Reviewer #3: **Yes: **Wisit Kaewput

---

## [Editor Report · Acceptance letter]

10 Mar 2022

PONE-D-21-39037R1 

Risk factors for Encapsulating Peritoneal Sclerosis in patients undergoing peritoneal dialysis: a meta-analysis 

Dear Dr. Wu:

I'm pleased to inform you that your manuscript has been deemed suitable for publication in PLOS ONE. Congratulations! Your manuscript is now with our production department. 

Kind regards, 

on behalf of

Dr. Wisit Cheungpasitporn 

Academic Editor

PLOS ONE